# An assessment of the fixin tplo jig to generate effective compression using a transverse fracture model

**Cassio Ricardo Auada Ferringo**[1]*, **George Diggs**[2], **Daniel D. Lewis**[3], **Scott A. Banks**[2]

**1** Department of Small Animal Clinical Sciences, College of Veterinary Medicine, University of Tennessee, Knoxville, TN, United States of America, **2** Department of Mechanical and Aerospace Engineering, College of Engineering, University of Florida, Gainesville, FL United States of America, **3** Department of Small Animal Clinical Sciences, College of Veterinary Medicine, University of Florida, Gainesville, FL, United States of America

* cferrign@utk.edu

**Data Availability Statement:** All relevant data are within the paper and its Supporting Information files.

**Funding:** The University of Florida, College of Veterinary Medicine's Mark S. Bloomberg

## Abstract

The objective of this study was to determine compressive loads that could be generated using a tibial plateau leveling osteotomy (TPLO) jig with a tensioned strand of 18-gauge stainless steel orthopedic wire in a simulated transverse fracture model. The wire was sequentially tensioned using heavy needle holders or an AO wire tightener. Recorded loads were subsequently compared to loads generated by applying a 3.5 mm limited contact-dynamic compression plate (LC-DCP) as a compression plate. Two segments of 2 cm diameter Delrin rod were placed in a testing apparatus and used to simulate a transverse fracture. A load cell was interposed between the two segments to measure the compressive loads generated during the application of the TPLO jig or the LC-DCP. Compression was generated by sequential tensioning a strand of 18-gauge wire secured through the base of the arms of the TPLO jig or by placing one or two load screws in the LC-DCP. Wires were tensioned using heavy needle holders or an AO wire tightener. Eight replicates of each construct were tested. Recorded loads were compared using a one-way repeated measures ANOVA and Tukey Honestly Significant Difference test. The wire being tensioned broke while attempting a second quarter rotation of the needle holders and when the crank handle of the AO wire tightener was advanced beyond two rotations. The mean + SD peak compressive loads recorded when tensioning the wire using the heavy needle holders and AO wire tightener was 148 ± 7 N and 217 ± 16 N, respectfully. The mean ± SD load recorded after placement of the first and second load screw in the LC-DCP was 131 ± 39 N and 296 ± 49 N, respectively. The compression generated by placing two load screws in the LC-DCP was superior to the compression generated using the jig. The maximum load recorded by tensioning the wire secured through the TPLO jig using the AO wire tightener was superior to the compression generated by placing a single load screw and tensioning the wire using needle holders. Our results demonstrate that the TPLO jig allows surgeons to compress transverse fractures or osteotomies effectively. Tensioning the AO wire tightener allows for sequential tensioning and generates superior compressive loads than tensioning wires with heavy needle holders.

Memorial Small Animal Surgery Resident Research Fund partially funded the project. Funding helped to pay for the load cell. The funders had no role in study design, data collection, and analysis, the decision to publish, or preparation of the manuscript.

**Competing interests:** The authors have declared that no competing interests exist.

## Introduction

Bone plates are often utilized to stabilize transverse fractures or osteotomies. Various intrinsic and extrinsic mechanisms have been developed to generate interfragmentary compression at the fracture or osteotomy site during plate application [1, 2]. Applied compression enhances stability by generating friction between the ends of the apposed osseous segments [1]. Anatomic reduction and interfragmentary compression can result in absolute stability, limiting interfragmentary strain to <2%, which promotes primary bone healing and mitigates the potential for implant failure by enhancing load sharing between the implants and the secured bone segments [1, 3–8].

Traditional plating systems, developed in the 1960s by AO (Arbeitsgemeinschaft für Osteosynthesefragen), are designed with an intrinsic mechanism to generate interfragmentary compression [1]. Dynamic compression plates (DCPs), limited contact dynamic compression plates (LC-DCPs), and even locking compression plates (LCPs) have an inclined plane located at one or both axial margin(s) of the screw holes. Appropriate eccentric screw placement results in the head of the screw interacting with the inclined plane, which invokes linear translation of the screw and the secured bone segment as the screw is tightened, generating compression at the fracture or osteotomy site [9]. Some newer locking plate systems marketed for veterinary applications do not have an intrinsic mechanism to generate interfragmentary compression [2, 10–12]. Massimo & Nicetto described the use of a tibial plateau leveling osteotomy (TPLO) jig (VS901/900, Intrauma SpA, Rivoli, Italy) to generate interfragmentary compression of a reduced transverse osteotomy when using a plating system without intrinsic compression capabilities [13]. This jig has a hole at the base of each of the device's articulating arms; these holes are designed to accept a strand of orthopedic wire. The jig can be secured to the major proximal and distal bone segments using half-pins, and a strand of wire can be passed through the holes located in the base of the articulating arms [13]. A set screw is inserted in an adjoining hole in the base of one of the articulating arms. As the set screw is tightened, the screw impinges on the wire and securing the wire in place. Tension can be applied to the segment of the wire protruding through the hole in the base of the other articulating arm. Once the wire has been tensioned, a second set screw is placed and tightened to maintain the tension in the wire. This process purportedly generates effective interfragmentary compression [13].

The objective of this study was to measure the compressive load generated by using an TPLO jig with a tensioned strand of 18-gauge stainless steel orthopedic wire in a simulated transverse fracture model. Two wire tensioning methods were evaluated: tensioning the wire using a pair of heavy needle holders or an AO wire tightener. We utilized the same transverse fracture model and measured the compression obtained by applying a 3.5 mm LC-DCP plate in compression to establish a comparative reference.

## Materials and methods

The testing apparatus (Fig 1) was designed using Solidworks (Dassault Systems, Waltham MA) and 3D printed in ABS plastic (Fortus 450MC, Stratasys, Edina, MN). The apparatus was designed to hold two 6 cm length segments of 2 cm diameter Delrin rod to simulate a mid-diaphyseal transverse long bone fracture. The ends of the rods were sanded with fine-grit sandpaper to remove irregularities that could cause spurious point loading. A 4mm diameter circular load cell (LC302-250, Omega) was positioned between the two Delrin rod segments.

The load cell is designed to measure central compression and not loads incurred peripherally.

The apparatus positioned the Delrin rod segments in a vertical orientation and allowed unencumbered linear translation of the segments during testing. Two versions of the

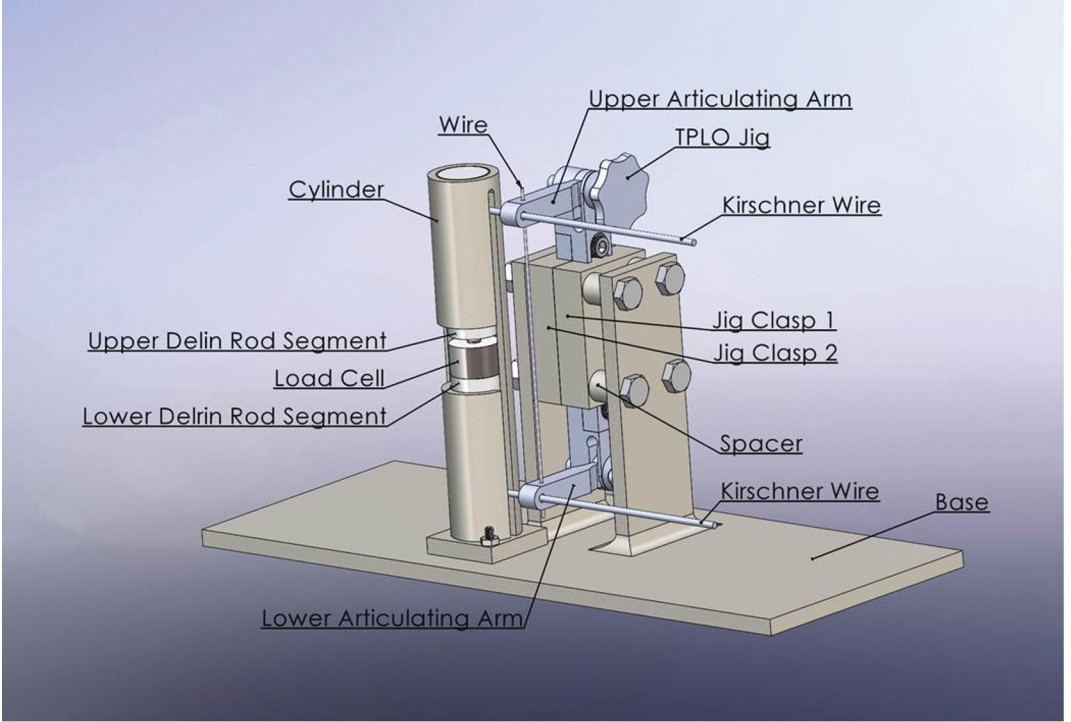

**Fig 1. Rendering of the testing jig.** A load cell was positioned between the two segments of the Delrin rod, which were secured within the testing apparatus' cylinder.

mounting apparatus were fabricated. The only variation between the two apparatuses was the width of the longitudinal open section in the cylindrical portion of the apparatus that secured the Delrin rod segments and load cell was appropriate to accommodate application of fixation pins or a bone plate.

The load cell was connected electronically to a solid-state analog amplifier (gain of 64) and 14-bit differential-input data acquisition device (SADI DAQ, Out of the Box Robotics, Gainesville, FL). LabView (National Instruments, Austin, TX) programming was used to record the differential amplifier output voltage ($V_{amp}$), the load cell excitation voltage ($V_s$), and the applied load using the relation:

$$F = \frac{V_{amp} * Max\ Load}{Sensitivity * V_s * AF}$$

Where F is the measured load (N), $V_{amp}$ is the amplifier output voltage (mV), *MaxLoad* is the calibrated maximum sensor load (1112 N), *Sensitivity* is the sensor sensitivity (0.934 mV/V), $V_s$ is the measured sensor excitation voltage (V), and AF is the amplification factor (64).

## TPLO jig with tensioning of the wire done using heavy needle holders

The Delrin rods were mounted horizontally into a tabletop vice, and a transverse hole was drilled through each segment of the rod using a 2.0 mm drill bit. The hole in what would become the proximal Delrin rod was located 13.5 mm from the distal end of the rod, which would be in contact with the load cell. The hole in what would become the distal Delrin rod was located 16 mm from the proximal end rod, which also would be in contact with the load cell. A negative profile 2.5 mm half-pin was inserted in each of these holes until the trocar tip

of the pin emerged through the trans-surface of the rod. The pins were withdrawn until the tip of the pins no longer protruded through the rod to avoid impingement on the testing apparatus cylinder during mechanical testing. The Delrin rods were slid into the mounting jig cylinder with the load cell positioned between the two rod segments with the half-pins protruding through the open section of the cylinder.

The half-pins were inserted horizontally through the orifices located remotely on the articulating arms of the TPLO jig and secured in place by tightening adjoining set screws with an Allen wrench. A 200 mm strand of 18-gauge orthopedic wire (IMEX Veterinary, Inc. Longview, TX) was fed through the orifices located at the base of each articulating arm (Fig 1). The wire was secured in the distal, dependent articulating arm by tightening the set screw with an Allen wrench. The corresponding set screw on the upper proximal articulating arm was removed, which allowed the wire to translate through the proximal articulating arm freely. Traction applied to the free end of the wire would pull the secured dependent, distal articulating arm upwards, resulting in proximal linear translation of the distal, dependent Delrin segment.

The segment of wire protruding from the hole in the upper, proximal arm of the jig was secured in the jaws of a pair of heavy needle holders. The ratchet mechanism of the needle holder was locked, the load cell and the Delrin rod were preloaded to 5 to 10 N, and the system recording amplified voltage was tared. Tensioning of the wire was performed by rotating the needle holders in a clockwise direction using the upper articulating arm as a fulcrum (Fig 2). The wire was progressively coiled around the jaws of the instrument. After each quarter

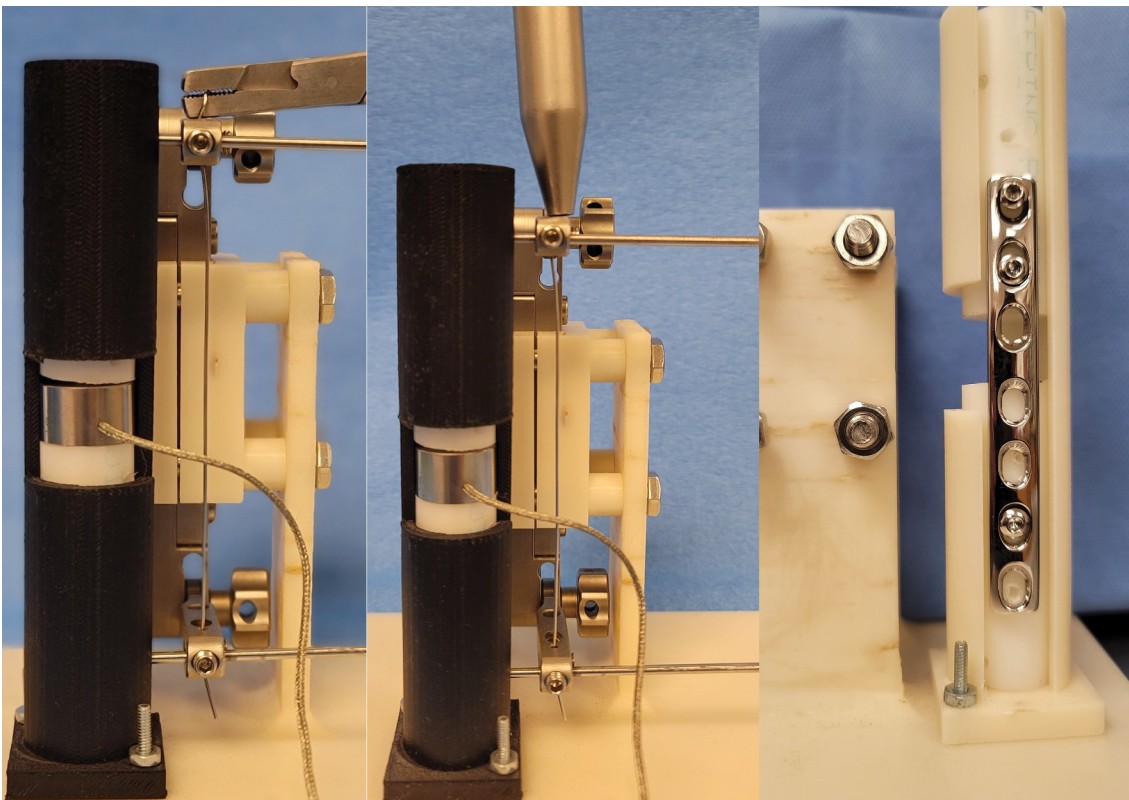

**Fig 2. Photographs of the experimental setup.** Top compression is being generated between the two Delrin rod segments by tensioning the wire secured through the TPLO jig using (A) heavy needle holders, (B) an AO wire tightener, or (C) application of a compression plate.

rotation, the position was held. This allowed the force applied to the load cell to settle. The needle holders were subsequently rotated in a clockwise direction, a quarter revolution at a time until the wire broke. The system was inspected for damage, and components were replaced as needed. The segment of wire was replaced, and testing was repeated eight times.

### TPLO jig with tensioning of the wire done using a wire tightener

After applying the TPLO jig and placing the orthopedic wire in a similar fashion as described previously, the free end of the wire was fed through the hole located in the nose of an AO wire tightener (DePuy-Synthes, Warsaw IN) and then through the cannulation in the instrument's crank handle (Fig 2). The crank handle was secured in the recesses of the tensioning device positioned closest to the nose of the device. The nose of the instrument was placed in contact with the proximal articulating arm of the TPLO jig as the crank handle was rotated a quarter turn. The wire protruding from the cannulation in the crank handle was trimmed 1 cm from the instrument. A preload of 5 to 10 N was obtained, and the system recording amplified voltage was tared. The wire was manually tensioned by rotating the crank handle in a clockwise direction in half revolution increments. After each half rotation of the crank handle, the position was held, allowing the load to settle. The crank handle was rotated in successive half revolution turns until the wire broke. After each test, the system was inspected for damage, and components were replaced as needed. The segment of wire was replaced, and testing was repeated eight times.

### Limited contact dynamic compression plating (LC-DCP)

A seven-hole, 3.5 mm, stainless steel LC-DCP was applied and compression was generated by sequentially placing two 3.5 mm screws in the load position (Fig 2). The ends of the Delrin rod segments were prepared as previously described, and the segments were placed into the testing apparatus cylinder with the load cell positioned between the segments of the rod. The plate was then placed on the surface of the rods exposed through the open section of the cylinder with the plate's central screw hole positioned directly over the load cell. Using a neutral drill guide, a 2.5 mm drill bit was used to drill a hole in the Delrin rod through the third most distal hole in the plate. A 3.5 mm cortical screw was inserted, affixing the plate to the dependent, distal Delrin rod. The proximal segment of the Delrin rod was allowed to contact the load cell, establishing a preload of 5–10 N, and a 2.5 mm drill bit was used to drill a hole in the Delrin rod in second-most proximal and second most distal holes in the plate using a load drill guide. 3.5 mm cortical screws were inserted into these holes but not fully tightened, as rotation of the screwdriver was stopped just prior to the head of the screws coming into contact with the plate.

The screw in the proximal Delrin segment was fully tightened, allowing five to 10 seconds to elapse after tightening before recording the load. The screw placed using the neutral drill guide was then loosened slightly before tightening the screw placed in the distal Delrin segment using the load drill guide. After tightening the second loaded screw, the load was allowed to plateau before recording the load. The construct was disassembled, and the process was repeated eight times.

### Statistical analysis

A one-way repeated measures ANOVA was used to compare measured loads across the seven conditions. A Tukey Honestly Significant Difference test examined pair-wise differences between conditions. Statistical significance was set at a p-value $<0.05$.

**Table 1. Compression loads generated between Delrin rod 'bone segments' for seven loading conditions.** The top panel reports mean and standard deviation values for eight evaluations of each loading condition. The lower panel reports the p-values for tests of pair-wise differences using Tukey's Honestly Significant Difference, where values < 0.05 are considered significant.

|  | NH 0.25 | WT 0.5 | WT 1.0 | WT 1.5 | WT 2.0 | LC-DCP 1 | LC-DCP 2 |
|---|---|---|---|---|---|---|---|
| Mean (N) | 148 | 106 | 166 | 198 | 217 | 131 | 296 |
| STD (N) | 7 | 34 | 15 | 14 | 16 | 39 | 49 |
|  | NH 0.25 | WT 0.5 | WT 1.0 | WT 1.5 | WT 2.0 | LC-DCP 1 | LC-DCP 2 |
| NH 0.25 |  | 0.077 | 0.883 | 0.018 | 0.001 | 0.896 | 0.001 |
| WT 0.5 | 0.077 |  | 0.003 | 0.001 | 0.001 | 0.599 | 0.001 |
| WT 1.0 | 0.883 | 0.003 |  | 0.291 | 0.015 | 0.223 | 0.001 |
| WT 1.5 | 0.018 | 0.001 | 0.291 |  | 0.840 | 0.001 | 0.001 |
| WT 2.0 | 0.001 | 0.001 | 0.015 | 0.840 |  | 0.001 | 0.001 |
| LC-DCP 1 | 0.896 | 0.599 | 0.223 | 0.001 | 0.001 |  | 0.001 |
| LC-DCP 2 | 0.001 | 0.001 | 0.001 | 0.001 | 0.001 | 0.001 |  |

## Results

In the constructs in which the needle holders were used to tension the wire, the wire broke adjacent to the jaws of the needle holder when attempting a second quarter rotation of the needle holders in all eight constructs. The mean ± SD compressive load recorded after the first quarter-rotation of the needle holders was 148 ± 7 N (Table 1).

In the constructs in which the AO wire tightener was used to tension the wire, the wire broke at the base of the crank handle while attempting to advance the crank handle two and a half rotations in all eight constructs. The mean ± SD loads measured after each half revolution of the crank handle were 106 ± 34 N, 166 ± 15 N, 198 ± 14 N, and 217 ± 16 N (Fig 3).

In the constructs in which the LC-DCP was applied, the mean ± SD load measured after tightening the first load screw in the LC-DCP was 131 ± 39 N. The mean ± SD load measured after tightening the second load screw was 296 ± 49 N (Fig 3). Failure did not occur in the plated constructs.

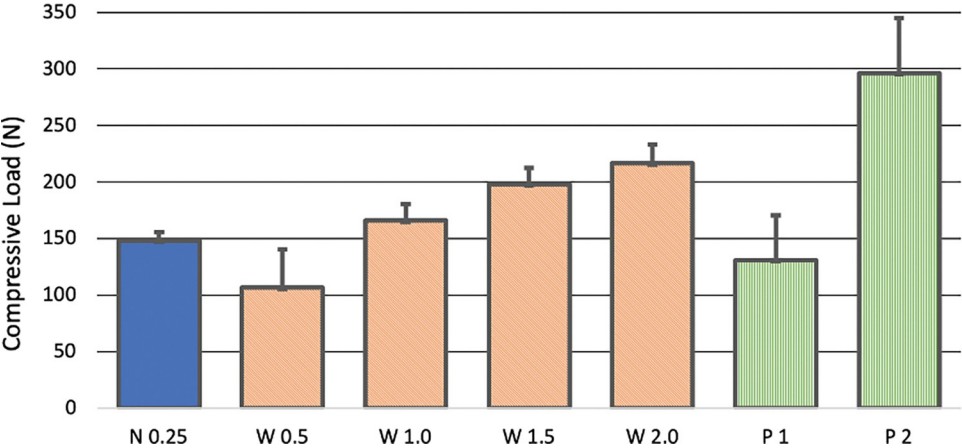

**Fig 3. Graphic showing the compression load (N) for each experimental group.** Mean ± SD loads measured for each of the seven conditions evaluated. Compression was generated by tensioning the wire secured in the TPLO jig using heavy needle holders and performing a quarter rotation ("blue", N 0.25), or an AO wire tightener and performing a half, one, one and a half, and two full rotations ("red", W 0.5, W 1.0, W 1.5, W 2.0), or by applying a LC-DCP with one or two screws placed in the load position ("green", P 1, P 2).

The one-way ANOVA showed there were significant differences (p < 0.001) across the seven loading conditions (Table 1). Pair-wise differences were not observed between compressive loads generated after a quarter rotation of the needle holders and a half and a full rotation of the crank handle of the wire tightener or tightening the initial load screw of the plate construct. One and a half or two turns of the crank handle of the wire tightener or two screws tightened on the compression plate generated higher compressive loads than a quarter-turn of the needle holders. Tightening the second load screw generated a greater compressive load than the other six conditions.

## Discussion

Our results substantiate that a TPLO jig and tensioned orthopedic wire can effectively compress a transverse fracture or osteotomy. Our methodology utilized a circular load cell positioned between two segments of Delrin rod, allowing objective compression of the compressive load generated by the seven application techniques assessed. The 3D printed custom testing apparatus allowed for stable linear translation of the Delrin rods, minimizing angular displacement, during wire tensioning and tightening of the screws securing the LC-DCP.

The Delrin rod was chosen for its more homogeneous material compared to cadaveric bone. Naturally occurring bone can introduce variability, such as differences in size, density, and shape that could impact the overall data collection in a biomechanical study. The study found that Delrin was a suitable material, which is consistent with other publications [14–16].

We were able to generate a maximum mean ± SD compressive load of 147 ± 7 N when tensioning the wire with the needle holders. The wire consistently broke adjacent to the jaws of the instrument when attempting to rotate the needle holders further after a quarter turn. The AO wire tightener generated greater compression and allowed for a more precise, graduated application of tension than the needle holders. The maximum mean ± SD compressive load generated using the AO wire tightener, 217 ± 16 N, was significantly greater than that obtainable when using the needle holders. The AO wire tightener coils the wire around a cylindrical mandrel, providing superior tensioning mechanics compared to the use of needle holders. Coiling the wire around the rectangular jaws concentrates strain at the instrument's squared external corners predisposing to wire failure at lower loads. In a biomechanical study comparing loads generated by various wire tensioning devices, loop knot wires tensioned with a wire tightener device similar to the AO wire tightener used in the current study produced greater final wire tension than twist knot wires, including wires tensioned with jawed instruments similar to the needle holders used in the current study [17]. Devices which coil a wire around a cylindrical mandrel deform the wire uniformly, in contrast to the concentrated deformation which occurs when a wire is tensioned using a rectangular jawed instrument that has multiple acute corners which serve as stress risers during tensioning [15]. Concentrated acute deformation of a wire causes the wire to break at lower loads [15], which is what occurred in the current study when the wire was tensioned with heavy needle holders.

The compressive load recorded after tensioning the wire in the TPLO jig after a quarter-rotation of the needle holder, or a single complete rotation of the crank of the wire tightener was not significantly different from the compression recorded after tightening the initial load screw in the LC-DCP. Tightening the second load screw significantly increased the compression afforded by the LC-DCP and the mean compressive load recorded was 36% greater than the maximum mean compressive load recorded in the jig construct in which the wire was tensioned with the AO wire tightener. Interfragmentary compression of anatomically reduced fracture segments increases friction between the apposed surfaces of the secured bone

segments, which enhances stability and protects an applied plate from bending forces [2, 4, 8, 18–20]. While optimal compressive loads have not been definitely established, one study performed using the AO articulated compression device stated that > 1000 N of compression was required to obtain an absolute rigidity [21]. In an early study performed using DCPs instrumented with a strain gauge mounted to the surface of the plate positioned directly over the osteotomy, initial loads of 700–1,800 N were reported after placing a single screw with the load drill guide [22]. The loads recorded in our study, even after placing a second screw with the load drill guide, are considerably lower. These discrepancies may reflect differences in methodology between studies. The relative differences in compressive loads recorded in the current study are probably more salient than the precise numeric figures recorded.

There are several limitations to this study. First, we only used 18-gauge orthopedic wire to generate compression with the TPLO jig, and we can assume higher compressive loads could have been obtained if we had used a larger diameter wire (Wilson JAVMA 1985, Neat Vet Surg 2006, Neat VCOT 2006). We speculate that compression comparable to that generated by inserting two screws in the LC-DCP using the load drill guide could be obtained using heavier gauge orthopedic wire and the AO wire tightener. One study showed that increasing wire diameter by 50% resulted in a 169% increase in load-to-failure, and doubling the wire diameter from 0.45 to 0.98 mm increased the load to failure by more than 300% [23]. Second, our method of manually tensioning the wire was subjective and not calibrated. Subjective wire tensioning has been utilized in prior biomechanical studies (Wilson JAVMA 1985, Willer VCOT 1990, Roe Vet Surg 1997, add the new paper by Roe I sent last week) and mimics what is done in actual clinical situations. This method yielded relatively consistent results, with standard deviations ranging from 7–16 N, and a relative standard deviation of less than 5%. The relative standard deviation of the loads recorded when tensioning wires with the heavy needle holder and the AO wire tightener were reasonably similar, 4.9% and 8.8%, respectively, demonstrating the consistency of application. The loads recorded when placing one or both load screws in the LC-DCP were more variable.

Our results confirm that the TPLO jig used in this study can be used to effectively compress transverse fractures or osteotomies when applying plating systems that do not have an inherent mechanism to generate interfragmentary compression. We recommend using a wire tightener for this purpose, as this instrument produced greater interfragmentary compression and can be applied more precisely than using a needle holder to tension the wire.

This information is relevant because some new plate system neutral locking plates in veterinary orthopedics is becoming increasingly popular. Thus, using the TPLO jig has the potential to compress the fracture gap using a neutral implant. In some fractures, if a stronger compression is desired, the jig could be used. Additionally, in a MIPO scenario where achieving good compression in the fracture gap is not always possible, this concept could also be utilized.

## Supporting information

**S1 File.**
(PDF)

**S2 File.**
(PDF)

**S3 File.**
(PDF)

## Author Contributions

**Conceptualization:** Cassio Ricardo Auada Ferringo, George Diggs, Daniel D. Lewis, Scott A. Banks.

**Data curation:** George Diggs, Scott A. Banks.

**Formal analysis:** George Diggs, Scott A. Banks.

**Funding acquisition:** Daniel D. Lewis.

**Investigation:** Cassio Ricardo Auada Ferringo, George Diggs, Daniel D. Lewis.

**Methodology:** Cassio Ricardo Auada Ferringo, George Diggs, Daniel D. Lewis.

**Project administration:** Cassio Ricardo Auada Ferringo, Daniel D. Lewis.

**Resources:** Daniel D. Lewis.

**Software:** George Diggs, Scott A. Banks.

**Supervision:** Cassio Ricardo Auada Ferringo, Daniel D. Lewis, Scott A. Banks.

**Validation:** Cassio Ricardo Auada Ferringo, George Diggs, Scott A. Banks.

**Visualization:** Cassio Ricardo Auada Ferringo.

**Writing – original draft:** Cassio Ricardo Auada Ferringo.

**Writing – review & editing:** Cassio Ricardo Auada Ferringo, George Diggs, Daniel D. Lewis, Scott A. Banks.

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
