## [Decision Letter · Decision Letter 0]

26 Jul 2023

PONE-D-23-15995An assessment of the fixin tplo jig to generate effective compression using a transverse fracture modelPLOS ONE

Dear Dr. ferrigno,

Thank you for submitting your manuscript to PLOS ONE. After careful consideration, we feel that it has merit but does not fully meet PLOS ONE’s publication criteria as it currently stands. Therefore, we invite you to submit a revised version of the manuscript that addresses the points raised during the review process.

We look forward to receiving your revised manuscript.

Kind regards,

Mohamed El-Sayed Abdel-Wanis, Ph.D.

Academic Editor

PLOS ONE

“yes

This study was supported by the University of Florida, College of Veterinary Medicine’s Mark S. Bloomberg Memorial Small Animal Surgery Resident Research Fund.”

4. Please include a copy of Table 1 which you refer to in your text on page 10.

5. We note that Figures 1 and 2 in your submission contain copyrighted images. All PLOS content is published under the Creative Commons Attribution License (CC BY 4.0), which means that the manuscript, images, and Supporting Information files will be freely available online, and any third party is permitted to access, download, copy, distribute, and use these materials in any way, even commercially, with proper attribution. For more information, see our copyright guidelines: http://journals.plos.org/plosone/s/licenses-and-copyright.

1. You may seek permission from the original copyright holder of Figures 1 and 2 to publish the content specifically under the CC BY 4.0 license.

Reviewers' comments:

Reviewer's Responses to Questions

**Comments to the Author**

1. Is the manuscript technically sound, and do the data support the conclusions?

Reviewer #1: Yes

Reviewer #2: Yes

2. Has the statistical analysis been performed appropriately and rigorously? 

Reviewer #1: No

Reviewer #2: N/A

3. Have the authors made all data underlying the findings in their manuscript fully available?

Reviewer #1: No

Reviewer #2: Yes

4. Is the manuscript presented in an intelligible fashion and written in standard English?

Reviewer #1: Yes

Reviewer #2: Yes

5. Review Comments to the Author

Reviewer #1: The authors demonstrate a biomechanical model comparing several surgical options for transverse fracture/osteotomy interfragmentary compression.

Methods and materials:

The 3D printed APS plastic was selected to simulate bone. Is it similar in its biomechanical properties to bone? If so provide evidence. There must be a rational why natural bones weren't used.

As an LCDCP plate utility is clear, it is not clear if the TPLO jig practical in clinical scenarios? In which exactly? Please provide an explanation, as not all of the readership of this journal come from the Veterinarian community.

The maintenance of compression over time is the end purpose of all fixation methods. Other than the read of compression force after 10 seconds there is no other reference to this issue. See this manuscript for further discussion:

Keltz E, Mora AJ, Wulsten D, Rußow G, Märdian S, Duda GN, Heyland M. Is initial interfragmentary compression made to last? An ovine bone in vitro study. Injury. 2021 Jun;52(6):1263-1270.

Statistical analysis:

It is not detailed enough.

A table of the individual values measured should be attached as supplementary material.

It is unclear what are the results that were analyzed with the Tukey Honestly Significant Difference test

Discussion

The clinical relevance of this model is not discussed enough.

Line 244: I would add that the main desirable effect by enhanced stability is not the protection of the plate but the reduction of shear forces which is the effect needed to ensure fracture union.

Limitations

Line 256-262: A fair point, but the correlation to practical real life clinical decision is missing. In which wire will the authors recommend to use in fracture surgery? If other than 18 gauge, why was this wire selected for their model? If not, what is their explanation?

Reviewer #2: The objective of this study was to determine compressive loads that could be generated using a tibial plateau leveling osteotomy (TPLO) jig with a tensioned strandof 18-gauge stainless steel orthopedic wire in a simulated transverse fracture model. The wire was sequentially tensioned using heavy needle holders or an AO wire tightener.

The results demonstrate that the TPLO jig allows surgeons to compress transverse fractures or osteotomies effectively. Tensioning the AO wire tightener allows for sequential tensioning and generates superior compressive loads than tensioning wires with heavy needle holders

6. PLOS authors have the option to publish the peer review history of their article (what does this mean?). If published, this will include your full peer review and any attached files.

Reviewer #1: No

Reviewer #2: No

---

## [Author Response · Author response to Decision Letter 0]

21 Aug 2023

ALL THE QUESTIONS ARE ANSWERED IN THE LETTER FROM THE EDITOR AND REVIEWER ONE, BUT I AM COPYING IT HERE.

This is the answer to reviewer one question and analysis response.

“The 3D printed APS plastic was selected to simulate bone. Is it similar in its biomechanical properties to the bone? If so provide evidence. There must be a rationale why natural bones weren't used.”

The Delrin rod was selected in the experimental environment as it is a more consistent, homogeneus material than bone. Natural bone introduces variability, such as size differences, bone density, and conformation that can affect the overall data collection in a biomechanical study.

On the other hand, several experiments have already been published that use Delrin rod as a reliable substitute for natural bone in biomechanical testing. I have included some references to papers that use a similar methodology with the Delrin rod..(1)(2)(3)(4)(5) 

“As an LCDCP plate utility is clear, it is not clear if the TPLO jig practical in clinical scenarios? In which exactly? Please provide an explanation, as not all of the readership of this journal come from the Veterinarian community.”

The use of neutral locking plates in veterinary orthopedics is becoming increasingly popular. However, many new generation implants lack the compress hole in the plate. Thus, using the TPLO jig has the potential to compress the fracture gap using a neutral implant. In some fractures, if a stronger compression is desired, the jig could be used. Additionally, in a MIPO scenario where achieving good compression in the fracture gap is not always possible, this concept could also be utilized.

“The maintenance of compression over time is the end purpose of all fixation methods. Other than the read of compression force after 10 seconds there is no other reference to this issue. See this manuscript for further discussion:

Keltz E, Mora AJ, Wulsten D, Rußow G, Märdian S, Duda GN, Heyland M. Is initial interfragmentary compression made to last? An ovine bone in vitro study. Injury. 2021 Jun;52(6):1263-1270.”

The manuscript cited by the first reviewer discusses the importance of maintaining compression over time in two different models of achieving compression in the fracture gap. The paper suggests that compression of more than 100 N might be necessary to overcome the loss of compression due to bone biomechanical characteristics and other physiological forces acting on the bone, such as axial displacement, in a real patient.

Our paper's scope differs from the previous one. While they measured if an implant could overcome compression loss over time, we focus on the amount of compression various techniques cause in a transverse fracture gap, aside from the conventional dynamic compression plate.

As mentioned in the paper cited by the reviser, it may be necessary to use different methods to achieve greater compression in the fracture gap in order to sustain it over time. Our paper provides some possible ways to achieve this.

“Statistical analysis: It is not detailed enough.

A table of the individual values measured should be attached as supplementary material.

It is unclear what are the results that were analyzed with the Tukey Honestly Significant Difference test”

A supplementary material containing graphics with all values has been added to the Plos One website as other files. The print of the statistical data has also been included to enhance the appreciation of the statistical treatment.

“The clinical relevance of this model is not discussed enough”

I believe that the clinical relevance of the project is already expressed in lines 302-305 in our conclusions paragraph. 

“Line 244: I would add that the main desirable effect by enhanced stability is not the protection of the plate but the reduction of shear forces which is the effect needed to ensure fracture union.”

I respectfully disagree with this statement. If a transverse fracture model is managed with compression and a plate or other bone implant is used to protect the fracture gap, there will not be any shear forces present. In other words, the bending forces will be negated and counteracted. 

If there is shared loading between the implant and the bone column, the friction of the plate in the periosteum will be “protected”. This shared loading can only occur with a compressive fracture gap. However, if the compression is too weak, a high-strain environment with minimal movement will hinder primary callus formation.

“Line 256-262: A fair point, but the correlation to practical real life clinical decision is missing. In which wire will the authors recommend to use in fracture surgery? If other than 18 gauge, why was this wire selected for their model? If not, what is their explanation?”

The decision to use the 18 gauge wire in the study was empirical. Further studies need to be done to find if different gauges of wires will yield higher or lower compression before the wire fails.

---

## [Decision Letter · Decision Letter 1]

8 Sep 2023

PONE-D-23-15995R1An assessment of the fixin tplo jig to generate effective compression using a transverse fracture modelPLOS ONE

Dear Dr. ferrigno,

Thank you for submitting your manuscript to PLOS ONE. After careful consideration, we feel that it has merit but does not fully meet PLOS ONE’s publication criteria as it currently stands. Therefore, we invite you to submit a revised version of the manuscript that addresses the points raised during the review process.I agree that the authors addressed all the reviewer comments, however a lot of the points explained in their detailed reply should be included in summaryin  the 'Discussion" of the revised manuscript

We look forward to receiving your revised manuscript.

Kind regards,

Mohamed El-Sayed Abdel-Wanis, Ph.D.

Academic Editor

PLOS ONE

Reviewers' comments:

Reviewer's Responses to Questions

**Comments to the Author**

1. If the authors have adequately addressed your comments raised in a previous round of review and you feel that this manuscript is now acceptable for publication, you may indicate that here to bypass the “Comments to the Author” section, enter your conflict of interest statement in the “Confidential to Editor” section, and submit your "Accept" recommendation.

Reviewer #1: All comments have been addressed

2. Is the manuscript technically sound, and do the data support the conclusions?

Reviewer #1: Yes

3. Has the statistical analysis been performed appropriately and rigorously? 

Reviewer #1: Yes

4. Have the authors made all data underlying the findings in their manuscript fully available?

Reviewer #1: Yes

5. Is the manuscript presented in an intelligible fashion and written in standard English?

Reviewer #1: Yes

6. Review Comments to the Author

Reviewer #1: (No Response)

7. PLOS authors have the option to publish the peer review history of their article (what does this mean?). If published, this will include your full peer review and any attached files.

Reviewer #1: No

---

## [Author Response · Author response to Decision Letter 1]

26 Sep 2023

The editors' letter contains the response, but all the information has been added to the manuscript.

---

## [Editor Report · Decision Letter 2]

28 Sep 2023

An assessment of the fixin tplo jig to generate effective compression using a transverse fracture model

PONE-D-23-15995R2

Dear Dr. Cassio Ferrigno,

We’re pleased to inform you that your manuscript has been judged scientifically suitable for publication and will be formally accepted for publication once it meets all outstanding technical requirements.

Kind regards,

Mohamed El-Sayed Abdel-Wanis, Ph.D.

Academic Editor

PLOS ONE

---

## [Editor Report · Acceptance letter]

4 Oct 2023

PONE-D-23-15995R2 

An assessment of the fixin tplo jig to generate effective compression using a transverse fracture model 

Dear Dr. Ferringo:

I'm pleased to inform you that your manuscript has been deemed suitable for publication in PLOS ONE. Congratulations! Your manuscript is now with our production department. 

Kind regards, 

on behalf of

Prof. Dr Mohamed El-Sayed Abdel-Wanis 

Academic Editor

PLOS ONE